# The G119S Acetylcholinesterase (*Ace-1*) Target Site Mutation Confers Carbamate Resistance in the Major Malaria Vector *Anopheles gambiae* from Cameroon: A Challenge for the Coming IRS Implementation

**DOI:** 10.3390/genes10100790

**Published:** 2019-10-11

**Authors:** Emmanuel Elanga-Ndille, Lynda Nouage, Cyrille Ndo, Achille Binyang, Tatiane Assatse, Daniel Nguiffo-Nguete, Doumani Djonabaye, Helen Irving, Billy Tene-Fossog, Charles S. Wondji

**Affiliations:** 1Centre for Research in Infectious Diseases (CRID), P.O. BOX 13591 Yaoundé, Cameroon; lynda.djounkwa@crid-cam.net (L.N.); cyrille.ndo@crid-cam.net (C.N.); achille.binyang@crid-cam.net (A.B.); tatianeassatse@gmail.com (T.A.); daniel.nguiffo@crid-cam.net (D.N.-N.); doumani.djonabaye@crid.cam.net (D.D.); billy.tene@crid-cam.net (B.T.-F.); charles.wondji@lstmed.ac.uk (C.S.W.); 2Department of Animal Biology and Physiology, Faculty of Science, University of Yaoundé 1, P.O. Box 812 Yaoundé, Cameroon; 3Department of Biological Sciences, Faculty of Medicine and Pharmaceutical Sciences, University of Douala, P.O. Box 24157 Douala, Cameroon; 4Department of Animal Biology, Faculty of Science, University of Dschang, P.O. Box 67 Dschang, Cameroon; 5Vector Group, Liverpool School of Tropical Medicine, Pembroke Place, Liverpool L3 5QA, UK; helen.irving@lstmed.ac.uk

**Keywords:** *Ace-1 G119S* mutation, insecticide resistance, *Anopheles gambiae*, Cameroon, malaria

## Abstract

Growing resistance is reported to carbamate insecticides in malaria vectors in Cameroon. However, the contribution of acetylcholinesterase (*Ace-1*) to this resistance remains uncharacterised. Here, we established that the *G119S* mutation is driving resistance to carbamates in *Anopheles gambiae* populations from Cameroon. Insecticide bioassay on field-collected mosquitoes from Bankeng, a locality in southern Cameroon, showed high resistance to the carbamates bendiocarb (64.8% ± 3.5% mortality) and propoxur (55.71% ± 2.9%) but a full susceptibility to the organophosphate fenitrothion. The TaqMan genotyping of the *G119S* mutation in field-collected adults revealed the presence of this resistance allele (39%). A significant correlation was observed between the *Ace-1^R^* and carbamate resistance at allelic ((bendiocarb; odds ratio (OR) = 75.9; *p* < 0.0001) and (propoxur; OR = 1514; *p* < 0.0001)) and genotypic (homozygote resistant vs. homozygote susceptible (bendiocarb; OR = 120.8; *p* < 0.0001) and (propoxur; OR = 3277; *p* < 0.0001)) levels. Furthermore, the presence of the mutation was confirmed by sequencing an *Ace-1* portion flanking codon 119. The cloning of this fragment revealed a likely duplication of *Ace-1* in Cameroon as mosquitoes exhibited at least three distinct haplotypes. Phylogenetic analyses showed that the predominant *Ace-1^R^* allele is identical to that from West Africa suggesting a recent introduction of this allele in Central Africa from the West. The spread of this *Ace-1^R^* represents a serious challenge to future implementation of indoor residual spraying (IRS)-based interventions using carbamates or organophosphates in Cameroon.

## 1. Introduction

During the last decades, the fight against malaria disease made significant progress, halving malaria deaths and decreasing its incidence by over a third [1,2]. These significant outcomes have been mainly driven by the scale-up of insecticide-based vector control interventions, such as long-lasting insecticidal nets (LLINs) and indoor residual spraying (IRS) [1,3]. Out of the four recommended insecticide classes in public health, pyrethroids have been the insecticides of choice for both strategies [1,4]. Unfortunately, the intense use of these chemicals for public health and agricultural purposes has led to the development of insecticide resistance in malaria vectors [4]. This rapid expansion of pyrethroid resistance could reverse progress achieved in reducing malaria burden due to the significant reduction of the efficacy of LLINs [5]. In order to sustain the efficacy of IRS and maintain or recover the efficacy of pyrethroids for insecticide-treated nets (ITNs), the World Health Organization (WHO) recommends application of insecticides having different mode of action or temporal replacement by different insecticide classes [6].

Over the past few years, there has been an increasing interest in using carbamate (CMs) and organophosphates (OPs) for public health purposes as alternatives to pyrethroids [7]. Indeed, numerous studies conducted under semi field conditions in experimental huts have shown the effectiveness of CMs and OPs against pyrethroid-resistant *Anopheles gambiae* mosquitoes [7,8,9,10,11,12]. Furthermore, the beneficial effects of these insecticides while used for IRS have largely been reported in several African countries [13,14,15,16,17]. Encouraged by these interesting results and with financial and technical support primarily from the United States President’s Malaria Initiative (PMI)/United States Agency for International Development (USAID), since 2006, several African countries started introducing the use of carbamate or organophosphate-based IRS in their national vector control strategy [13,16,18,19,20,21]. Unfortunately, a reduced susceptibility to CMs has been increasingly observed in some *A. gambiae* populations from West Africa [22,23,24,25,26,27]. This reduced susceptibility is associated with the emergence of the G119S mutation in *Ace-1* gene of *A. gambiae* mosquito [22,25,26,28,29]. This mutation resulting from a single amino acid substitution at codon 119 from glycine to serine (G119S) was reported to confer cross-resistance to CMs and OPs in mosquito species [30,31].The spread of this mechanism of resistance represents a serious threat for the effectiveness of IRS implementation in Africa. In contrast, in Central Africa, resistance to CMs had so far only been moderate with little or no evidence that *Ace-1* was playing any role [32]. This has led the President’s Malaria Initiative (PMI) program, which was recently implemented in Cameroon, to include the use of carbamate- and organophosphate-based IRS as a core component of the malaria control strategy in Cameroon [33]. The implementation of this strategy is expected to improve vector control in this country where high pyrethroid resistance level have been reported in *Anopheles* mosquito species [34]. Nevertheless, the effectiveness of this strategy could be limited by the resistance to CMs already reported by some previous studies in *A. gambiae* populations of Cameroon [32,34,35,36,37]. To avoid a rapid loss of effectiveness of such IRS control intervention, it is important to evaluate the current level of resistance to these insecticide classes and also to assess the potential contribution of the G119S mutation particularly as it confers cross-resistance to both CMs and OPs. 

The present study characterised the mechanisms involved in the resistance to carbamate detected in *A. gambiae* population from southern Cameroon. The G119S *Ace-1* mutation was detected with significant correlation with carbamate resistance whereas evidence of duplication of the gene was found.

## 2. Methods

### 2.1. Mosquito Sampling

Adult and larval stages of *A. gambiae* s.l. mosquitoes were collected in the locality of Bankeng (4°38′43″ N; 12°13′03″ E), a recent irrigated rice growing village in forest area in central Cameroon, as part of a study on the impact of rice cultivation on malaria transmission. Adult female mosquitoes were collected indoor on the walls and on the roof of different houses across the village between 6:00 AM and 10:00 AM using electric aspirators (Rule In-Line Blowers, Model 240). Mosquitoes were kept in paper cups and transported to the insectary of the Centre for Research in Infectious Diseases (CRID) in Yaoundé where they were morphologically identified and sorted by species according to the morphological identification keys of Gillies and De Meillon [38] and Gillies and Coetzee [39]. Mosquitoes were thereafter stored at −80 °C for molecular analysis. Mosquitoes were collected at the larval stage from *A. gambiae* s.l. specific breeding sites across the village using the dipping method. Larvae from stages 1 to 4 and pupae were transferred in bottles and then transported to the insectary where they were reared until the adult stage.

### 2.2. Insecticide Bioassays

Insecticide bioassay tests were carried out using two- to five-day-old female adults obtained from field collected larvae. Unfed mosquitoes were exposed to 0.1% bendiocarb, 1.0% propoxur, and 1.0% fenitrothion-treated papers for one hour as well as to a control paper (carrier oil-impregnated) following WHO standard procedures [40]. A quality control of the insecticide-impregnated papers was assessed using the *A. gambiae* susceptible laboratory strain Kisumu. The mortality rates were recorded 24 h after exposure, and WHO criteria were used to determine the resistance status of mosquitoes. Alive mosquitoes after exposure were kept at −80 °C whereas dead individuals were stored in silica gel and kept at −20 °C.

### 2.3. Species Identification and Ace-1 G119S Mutation Genotyping

These analyses were done using total genomic DNA extracted from 91 field-collected adult mosquitoes randomly selected (F_0_) and F_1_ alive and dead mosquitoes after exposure to bendiocarb (25 alive and 67 dead) and propoxur (30 alive and 38 dead). DNA was extracted from whole mosquito following the Livak protocol previously described [41]. Identification of species within *A. gambiae* complex was determined using the Short INterspersed Elements (SINE) PCR protocol [42]. The presence of the G119S mutation was screened with TaqMan real-time PCR assay (using Agilent Mx3005 qRT-PCR thermocycler (Santa Clara, CA, USA) following the protocols established by Bass and colleagues [43]. Each reaction was conducted in a total volume of 10 µL comprising 5 µL SensiMix (Bioline, London, UK), 0.25 µL of 40x Probe Mix coupled to allelic-specific primers, 4.25  µL of dH_2_0, and 1 µL of genomic DNA. Thermocycling conditions were an initial 10 min at 95 °C, followed by 40 cycles each of 92 °C for 15 s and 60 °C for 1 min. Two probes labelled with fluorochromes FAM^TM^ and HEX^TM^ were utilised to detect the resistant mutant and the wild-type susceptible alleles, respectively. Genotypes were scored from bi-directional scatter plots of results produced by the Mx3005 v4.10 software (Agilent). Thereafter, the correlation between G119S genotypes and bendiocarb resistance phenotypes was assessed by estimating the odds ratio (OR) using Vassar stats (http://vassarstats.net/) with a 2 × 2 contingency table. In each case, the proportion of resistant genotype or allele was compared to the susceptible one and the statistical significance was estimated based on Fisher’s exact probability test.

### 2.4. Ace-1 Gene Amplification, Sequencing, and Cloning

A region of 924 bp in a sequence of the *Ace-1* gene, encompassing exons 4–6 (VectorBase AgamP3 annotation, AGAP001356; G119S position in exon 5 corresponding to the third coding exon) was amplified from 55 female *A. gambiae*: 15 from F_0_ (field-collected adult mosquitoes), 40 from F_1_ mosquitoes after exposure to insecticide (10 alive and 10 dead after exposure to bendiocarb, 10 alive and 10 dead after exposure to propoxur). The amplification by PCR was carried out following the protocol previously described by Essandoh and collaborators [25]. Briefly, each reaction was conducted on a total volume of 50 μL containing 10 picomoles of each primer Ex2Agdir1 (5′AGG TCA CGG TGA GTC CGTACG A 3′) and Ex4Agrev2 (5′ AGG GCG GAC AGC AGA TGC AGC GA 3′), 10 mM dNTPs, ddH_2_O, 5X HF Phusion buffer, and 1 u of Phusion Taq polymerase (Fermentas, Burlington, ON, Canada). The cycle parameters were one cycle at 98 °C for 4 min, followed by 35 cycles of 98 °C for 30 s, 64 °C for 15 s, and 72 °C for 30 s, with final extension at 72 °C for 5 min. The PCR products were purified using the Qiaquick purification kit (Qiagen, Hilden, Germany). Out of the 40 samples used, 28 successful amplified (12 F_0_ field collected adults, 8 alive and 8 dead after exposure to bendiocarb). These amplicons were sequenced directly using the primers Ex2Agdir1 and Ex4Agrev2 to confirm the presence of the G119S mutation and assess signature of selection at this *Ace-1* in this location.

To investigate the presence of *Ace-1* duplication, purified DNA amplified from 18 alive mosquitoes after exposure to bendiocarb (8 mosquitoes) and propoxur (10 mosquitoes) were selected for cloning using the CloneJET™ PCR Cloning Kit (Thermo scientific, Waltham, MA, USA). The colonies were screened for the presence of the inserted amplicon using the supplied pJET1.2 primers according to the manufacturer’s instructions, and bands of approximately 900 bp were regarded as potential the *Ace-1* clones. Thereafter, for each individual, five clones were amplified, purified, and sequenced. All the successfully sequenced samples were aligned using ClustalW [44] as implemented in Bioedit software [45]. The alignment was done with the consensus sequence from Kisumu strain exported from VectorBase (gene ID: AGAP001356). The polymorphism analysis was performed using DnaSP v5.10 [46], while MEGA 10.1.0 [47] was used to build a maximum-likelihood tree from the aligned sequences after equalization length using the Tamura 3 parameter model selected after performing the model test. A haplotype network was also constructed using TCS program [48] and tcsBU [49].

## 3. Results

### 3.1. Mosquito Collection and Species Molecular Identification

A total of 323 indoor resting blood-fed females (F_0_) were collected and were all morphologically identified as members of the *A. gambiae* complex. Out of the 200 F_0_ mosquitoes randomly selected and tested for molecular identification, 98.5% (198/200) were *A. gambiae*, whereas only two mosquitoes were identified as *Anopheles coluzzii*.

### 3.2. Insecticide Bioassay

Overall, 260 F_1_ female adult mosquitoes aged two to five days obtained from field-collected larvae were exposed to bendiocarb, propoxur, and fenitrothion. Resistance was detected for the two carbamates tested with mortality rates of 64.8% ± 3.5% and 55.71% ± 2.9%, respectively for bendiocarb and propoxur. However, exposure to fenitrothion led to a 100% mortality showing a full susceptibility to this insecticide (Figure 1) No mortality was reordered in control tubes. 

### 3.3. Ace-1 Mutation Genotyping and Association with Insecticide Resistance Profile

*Ace-1* mutation was genotyped in both F_0_ field-collected mosquitoes and F_1_ female mosquitoes exposed to insecticide. The *119S* resistant allele was detected in 38.7% (34 homozygotes and 2 heterozygotes) out of the 93 F_0_ field-collected mosquitoes randomly screened. Out of the 25 alive mosquitoes after exposure to bendiocarb, 76%, 8%, and 16% of alive mosquitoes were genotyped homozygote resistant (S/S), heterozygote (G/S), and homozygote susceptible (G/G genotype), respectively (Figure 2a, Appendix A). In contrast, for dead mosquitoes, 4.5% were S/S, 1.5% G/S, and 94% G/G. For propoxur, 100% of dead mosquitoes were homozygote susceptible whereas 96.6% and 3.4% of alive mosquitoes were homozygote resistant and homozygote susceptible, respectively (Figure 2b, Appendix A). The *Ace-1*^R^ mutation was strongly associated with carbamate resistance for both allelic (odds ratio [OR] = 75.90; 95% confidence interval [CI]: 18.72–307.8 for bendiocarb; OR = 1514; 95% CI: 59.5–38,560 for propoxur) and genotypic (OR = 120.8; 95% CI: 25.0–583.3 and OR = 3277; 95% CI: 130.2–82,490 for bendiocarb and propoxur, respectively) levels. 

### 3.4. Genetic Diversity of Ace-1 in Bankeng

A region of 924 bp including the 119 codon of the *Ace-1* gene was amplified from 28 mosquitoes (12 F_0_, 8 dead, and 8 alive after exposure to bendiocarb) in order to confirm the presence of the 119S allele and to assess the genetic diversity of this gene. A 705 bp sequence was commonly aligned for the 28 samples (Appendix A). A G-to-A substitution at position 397, corresponding to the 119 codon, was observed in 11 sequences (seven F_0_ and four F_1_ alive) in comparison with the reference sequence from susceptible Kisumu strain, (Figure 3). Heterozygote mosquitoes were detected (two F_0_ and two F_1_ alive mosquitoes) with overlapping peaks for G and A at the same position (represented by the ambiguity code R, Figure 3). Interestingly, no substitution was detected in all the sequences from the eight dead F_1_ mosquitoes (Figure 3).

Analysis of the polymorphism patterns of the *Ace-1* portion resulted in the alignment of a common 705 bp detecting overall 35 polymorphic sites with a higher value of 25 and 29 in alive and F_0_ populations respectively and lower value in dead (three) individuals (Table 1. The number of haplotypes, the haplotype diversity, and the genetic diversity were higher for F_0_ and F_1_ alive mosquitoes than those for F_1_ dead mosquitoes. Most substitutions were synonymous with only the G119S as the single non-synonymous substitution (Table 1). 

A total of 23 different haplotypes were identified including 4, 8, and 10 specific to dead, alive, and F_0_ mosquitoes respectively, while 1 haplotype (H13) is shared by dead and alive mosquitoes, one (H11) by dead and F_0_ and another one (H3) by alive and F_0_ mosquitoes (Figure 4). The analysis of the haplotype network showed that H3 and H11 were the dominant haplotypes. Furthermore, a trend of clustering was observed according to phenotype, with all susceptible ones grouped in one cluster and the resistant ones in another cluster (Figure 4b). The phylogenetic tree emphasised this observation by clearly showing specific cluster between resistant (F_0_ and F_1_ alive individuals genotyped as RR by TaqMan assay) and susceptible (F_1_ dead individuals genotyped as SS) mosquitoes (Figure 4c). Interestingly, the predominant resistant haplotype from F_0_ and F_1_ alive mosquitoes was identical to resistant alleles previously detected in Ghana (accession number: KP165343, NCBI database, [25]) and Togo (accession number: KM875636; NCBI database, [50]), in the West African region.

### 3.5. Investigation of Duplication of Ace-1 in Bankeng

In order to investigate the presence of the *Ace-1* duplication, the same *Ace-1* portion from F_1_ mosquitoes alive after exposure to insecticide was cloned. Out of the 10 samples successfully cloned and sent for sequencing, seven (three exposed to bendiocarb: BenA1, BenA4, BenA7 and four exposed to propoxur: PropA5, PropA8, PropA9, PropA10) were successfully sequenced and analysed (Figure 5, Appendix A). Overall, each of these samples provided a minimum of three cloned haplotypes useful for investigating the presence of duplications. Except for sample BenA4 which contained only a single resistant haplotype, most mosquitoes carried at least three different haplotypes. A single glycine allele (susceptible) was observed for each sample, whereas, two and three different serine allele (resistant) were detected in four (BenA1, PropA5, PropA8, PropA9) and two (BenA7 and PropA10) different mosquitoes (Figure 5a,c). The haplotype network shows two different clusters: One composed by resistant alleles and another by mostly susceptible allele (Figure 5b). The allele H6 was the major resistant haplotype whereas there is no dominant allele among susceptible alleles.

Furthermore, a joint analysis (haplotype networks and phylogenies) of the data used in Figure 4 and Figure 5 was performed to further clarify the evolutionary path that led to the emergence of resistance haplotypes combining duplications and 119S. For this purpose, a common region of 703 bp was analysed for the directly sequenced and cloned samples. This analysis led to the identification of 39 different haplotypes including 18 resistant and 21 susceptible (Appendix A). The new haplotype network (Figure 6b) as well as the phylogenetic tree (Figure 6c) showed a clear clustering between resistant and susceptible haplotypes. Interestingly, the phylogenetic tree shows a higher haplotype diversity for susceptible specimens, whereas this diversity was low among resistant mosquitoes (Figure 6c, Appendix A). Furthermore, it can be observed that resistant haplotypes from duplicated specimens are almost all strongly similar to those from non-duplicated specimens (Figure 6b,c). Despite the observed high diversity, susceptible haplotypes from duplicated specimens are mostly close to those from non-duplicated ones. However, a susceptible haplotype H13-d is nested within a resistant cluster at two mutational steps from the dominant resistant haplotype H4 suggesting a possible reversion to the wild type from a resistant haplotype.

## 4. Discussion

Encouraged by interesting results observed in the reduction of malaria transmission in countries where non-pyrethroid-based IRS has been intensively implemented during the last decade, several other countries in Africa are planning to start using this strategy to control malaria. Carbamate and organophosphates are the two insecticide classes mostly currently used for IRS in areas of high pyrethroids resistance. Unfortunately, resistance to these insecticides is now being reported in malaria vectors across the African continent. To preserve the efficacy of IRS it is essential to understand the mechanisms underlying this resistance. In Cameroon, where IRS is planned to be implemented shortly through PMI activities, resistance to carbamate has already been reported in *A. gambiae* mosquitoes [32,34,51]. However, up to now, the molecular mechanisms involved in this resistance have not been characterised. The present study showed the evidence of *Ace-1* mutation in *A. gambiae* mosquito population from Cameroon and its association with carbamate resistance. Moreover, the analysis of the sequence bearing the G119S mutation led to the detection the duplication of this mutation in carbamate-resistant mosquitoes.

A high level of carbamate resistance was observed in the *A. gambiae* population tested in the present study and is consistent with other previous studies across the country [32,35,36,37,51]. As the use of carbamate and organophosphate insecticides for public health has not been effective to date or is very limited in Cameroon, it could be assumed that the primary source of selection must be from agricultural usage. This hypothesis could be supported by previous results of Antonio-Nkondjio and collaborators showing that mosquitoes originating from cultivated sites were more resistant to bendiocarb than those collected elsewhere [32]. This can be reinforced by the presence of significant watermelon fields using a significant quantity of pesticide in the village where mosquito collection was carried out. Furthermore, agriculture-driven selection of resistance to carbamates in *A. gambiae* mosquitoes was abundantly reported in West Africa [24,25,26,52].

Cross-resistance to carbamates and organophosphates have been reported to be conferred by the *Ace-1* mutation (G119S) due to a substitution of glycine by the serine in codon 119 of the gene [30,31]. Results of the present study demonstrated the evidence of a strong association between resistance to carbamates and the presence of G119S mutation in *A. gambiae* mosquito population from southern Cameroon. Indeed, almost all alive mosquitoes after exposure to both bendiocarb and propoxur were either homozygote serine or heterozygote TaqMan genotyped. Furthermore, the replacement of the G by the A nucleotide, leading to substitution of the glycine by the serine, was identified in the sequences of *Ace-1* gene from alive mosquitoes but not in the sequence the dead mosquitoes. These results clearly demonstrate that the *Ace-1* mutation is significantly involved in the occurrence of resistance to carbamates in the *A. gambiae* population from Bankeng. In our knowledge, this is the first time the G119S *Ace-1* mutation is clearly shown to be associated with carbamate resistance in Central African *A. gambiae* mosquito populations. Previous studies reporting the resistance to carbamates in *A. gambiae* mosquito populations from Central African countries did not detect the presence of *Ace-1* G119S mutation in this region or did not establish such association [32,53,54,55,56,57].

The *Ace-1* G119S mutation has been largely reported in West Africa but not in Central Africa. Its recent emergence in Cameroon could be explained by either a de novo occurrence in local populations of *A. gambiae* or could result from a spread of this mutation from West African populations. The result of the present seems to favour the hypothesis of a migration, as the resistant allele detected here was found identical to those previously detected in Ghana and Togo [25] and in other West African countries [30,52]. Further studies are needed to fully establish the origin of this mutation in Cameroon. However, the high frequency of the resistance allele (119S) and high ratio of mutant homozygotes in all the screened individuals is largely surprising knowing that the mutation seems to be recent in the *A. gambiae* population from Cameroon. Such high allelic frequency and heterozygous deficit was reported to be resulting from a deviation from the Hardy–Weinberg equilibrium in previous studies in West Africa [24,29]. 

In the present study, the detection of at least three different alleles in some individuals after cloning of the portion of the gene provides the evidence of an *Ace-1* gene duplication occurrence in a field population of *A. gambiae* from Cameroon. This is interesting as it seems to indicate that the selection of the *Ace-1* G119S mutation and the occurrence of the duplication are two events taking place under the same selective pressure. According to the result of the joint analysis of a common region for the directly sequenced and cloned samples, it appears that the 119S mutation would have first occurred on a duplicated haplotype. However, further genetic studies would be more informative for the understanding of this phenomenon. A higher haplotype diversity was observed for susceptible specimens, whereas this diversity was low among resistant mosquitoes suggesting a selective sweep acting on *Ace-1* gene in carbamate resistant mosquitoes in this location. This is similar to signatures of selection observed for other resistance loci in *A. gambiae* both for target-site and metabolic resistance [58] as well as in *Anopheles funestus* for GST [59] and P450-based [60] metabolic resistance mechanism.

The presence of three or more *Ace-1* alleles in *A. gambiae* mosquito was previously documented in several countries in West Africa [25,61,62]. In the present study, each sequenced individual specimen possessed at least two distinct resistant alleles and one susceptible allele. This could also explain why most mosquitoes alive after carbamate exposure were genotyped as homozygote resistant by TaqMan with a lack of heterozygotes as mosquitoes with two copies of the gene seem to have three resistant alleles of vs. only one susceptible allele. This is also consistent with the result of Essandoh and collaborators in Ghana but is in contrast to previous findings in Burkina-Faso and Côte-d’Ivoire, where only one resistant and two susceptible allele were detected in *A. gambiae* mosquitoes [61]. It was reported that the presence of this duplication allows individuals to have both susceptible and resistant copies of the gene, which likely decreases fitness costs associated with the resistant genotype [63]. Thus, the presence of such mutation represents an important threat for carbamate-based vector control strategy because it could not only allow mosquitoes to survive in the presence of insecticide, but also to reduce the impact of fitness cost in absence of insecticide pressure. 

## 5. Conclusions

This study demonstrates the presence of G119S *Ace-1* mutation associated with resistance to carbamate insecticides in a field population of *A. gambiae* in Cameroon. Furthermore, it also detected a duplication of the *Ace-1* mutation that potentially maintains the carbamate resistance in field populations by reducing the associated fitness cost. The emergence and the spread of this mutation could widely impact the effectiveness of all strategy based on the use of carbamate insecticides. To ensure the effectiveness of the planned IRS in Cameroon, there is an urgent need to conduct further studies to assess the distribution of the *Ace-1* G119S mutation and its association with resistance nationwide. 

## Figures and Tables

**Figure 1 genes-10-00790-f001:**
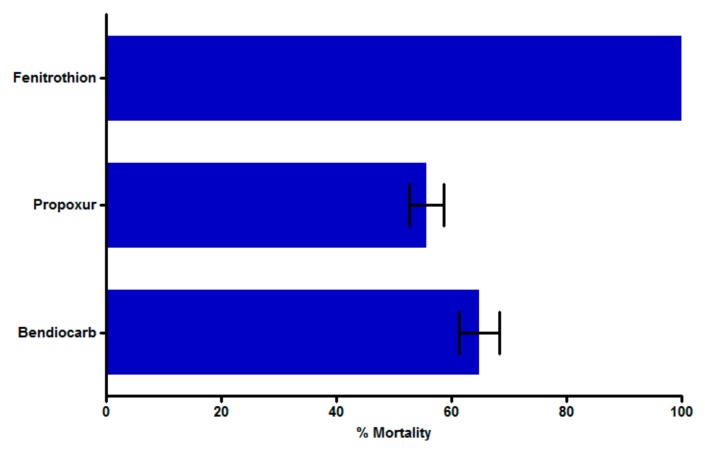
Susceptibility status of *Anopheles gambiae* mosquito population from Bankeng, central Cameroon. Mortality rates were recorded 24 h post-exposure to insecticides. Data are shown as mean ± standard error of the mean (SEM) (*n* = 260).

**Figure 2 genes-10-00790-f002:**
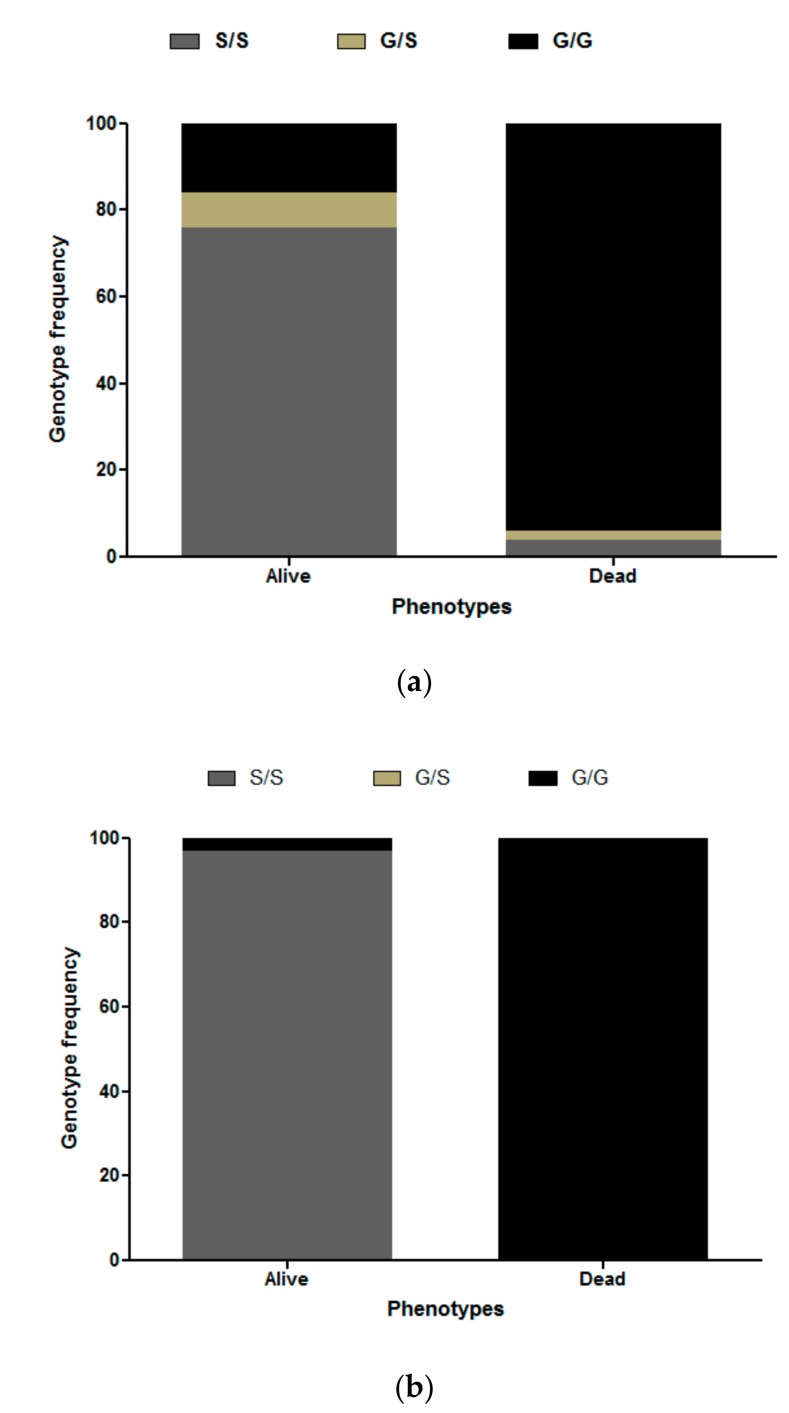
Distribution of *Ace-1* G119S genotypes and association with bendiocarb (**a**) and propoxur (**b**) resistant phenotype. Homozygote resistant (S/S), heterozygote (G/S), and homozygote susceptible (G/G genotype).

**Figure 3 genes-10-00790-f003:**
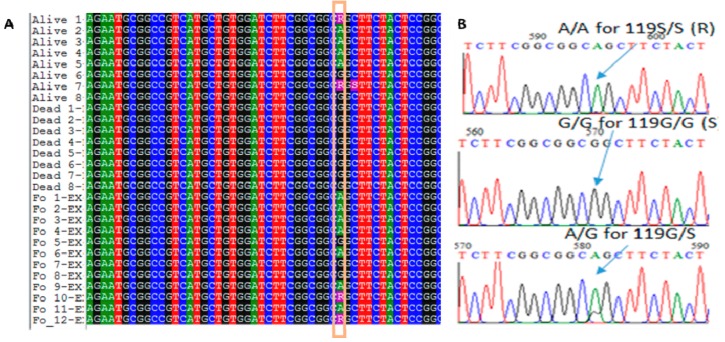
Sequencing of the portion of the *Ace-1* gene spanning the G119S mutation. (**A**) Sequence alignment of the *Ace-1* gene at the G119S point mutation in field collected adult mosquitoes (F_0_), F_1_ alive, and dead mosquitoes 24 h after exposure to bendiocarb. *R* represents the heterozygote genotype A/G. (**B**) Chromatogram traces showing the three genotypes at the 119 codon position.

**Figure 4 genes-10-00790-f004:**
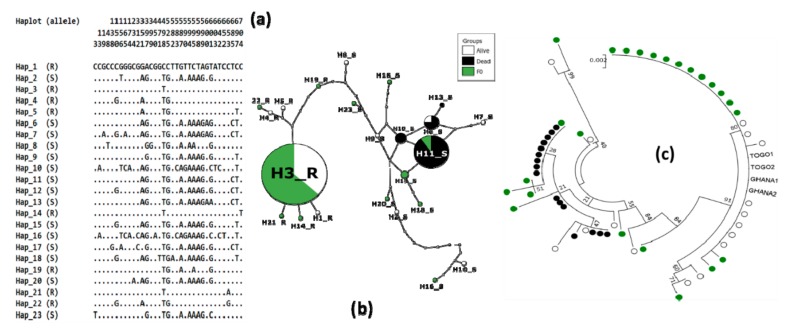
Polymorphism patterns of *Ace-1* gene from direct sequencing. (**a**) Polymorphic sites and haplotypes detected. Haplotypes are labeled with S (susceptible) or R (resistant). (**b**) The method of Templeton, Crandall and Sing (TCS) haplotype network showing the resistant and susceptible haplotype clusters. Lines connecting haplotypes and each node represent a single mutation event. (**c**) Maximum-likelihood phylogenetic tree of *Ace-1* gene supporting the clustering of haplotypes according to mosquito resistance status.

**Figure 5 genes-10-00790-f005:**
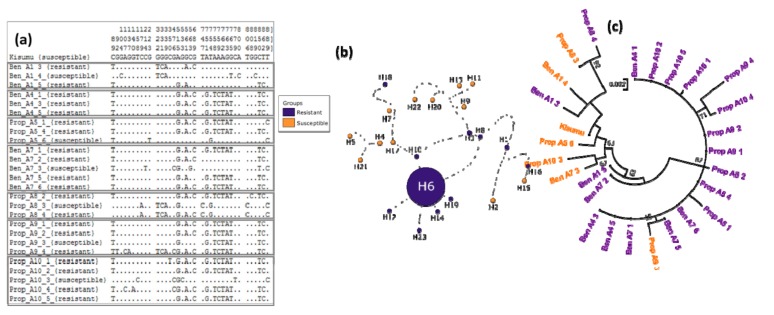
Polymorphism patterns of *Ace-1* gene from cloning. (**a**) Polymorphic sites and haplotypes detected. (**b**) TCS haplotype network showing the resistant and susceptible haplotype clusters. Lines connecting haplotypes and each node represent a single mutation event. (**c**) Maximum-likelihood phylogenetic tree of *Ace-1* gene supporting the clustering of haplotypes according to mosquito resistance status.

**Figure 6 genes-10-00790-f006:**
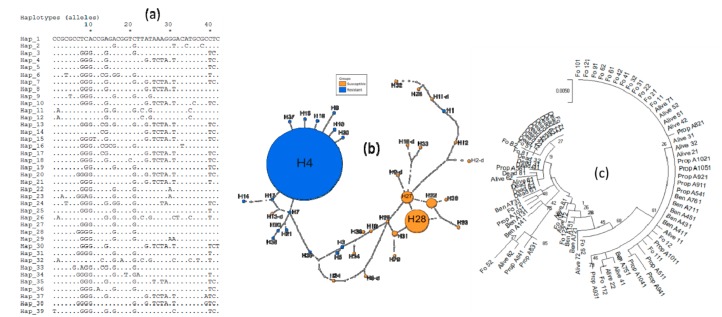
Polymorphism patterns of a common region of *Ace-1* gene from cloning and from direct sequencing. (**a**) Polymorphic sites and haplotypes detected. (**b**) TCS haplotype network showing the resistant and susceptible haplotype clusters. Lines connecting haplotypes and each node represent a single mutation event. The “d” at end indicates the susceptible haplotype from duplicated specimens. (**c**) Maximum-likelihood phylogenetic tree of *Ace-1* gene supporting the clustering of haplotypes according to the 119S genotypes.

**Table 1 genes-10-00790-t001:** Summary statistics for polymorphism in *Ace-1* gene including the G119S mutation in *A. gambiae* mosquito population from Bankeng, Central Cameroon.

	2n	S	Ka	Ks	h	hd	π	D	D*	Fs
**Alive**	16	25	1	8	10	0.825	0.01	−0.384 ns	−0.801 ns	0.561 ns
**Dead**	16	3	0	1	4	0.650	0.001	0.467 ns	−0.038 ns	−0.151 ns
**F_0_**	24	29	1	12	10	0.757	0.009	−0.755 ns	−1.721 ns	0.588 ns
**Total**	56	35	1	14	23	0.853	0.01	−0.507 ns	−2 ns	−3.695 *

2n: number of sequences; S: number of polymorphic sites; Ka: synonymous substitution; Ks: non-synonymous substitution; h: number of haplotypes; hd: haplotype diversity; π: nucleotide diversity; D: Tajima’s statistics; D*: Fu and Li’s statistics (the asterisk indicates “without an outgroup”); Fs: differences between sequences; ns: Not significant.

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
