# Peer review of "The G119S Acetylcholinesterase (Ace-1) Target Site Mutation Confers Carbamate Resistance in the Major Malaria Vector Anopheles gambiae from Cameroon: A Challenge for the Coming IRS Implementation"

_genes, 2019, doi:10.3390/genes10100790_

Round 1
Reviewer 1 Report
This article “The G119S acetylcholinesterase (Ace-1) target site 2 mutation confers carbamate resistance in the major 3 malaria vector Anopheles gambiae from Cameroon: A challenge for the coming IRS implementation” is an extremely interesting research paper describing the actual situation of insecticide resistance in Cameroun which may jeopardize the current vector control strategies. The study was well designed and performed. The authors give a good historical overview on carbamate and organophosphates insecticides resistance in west and Central African and how it may select or migrate in Cameroun.
Moreover, the present article will be of highest interest not only for Cameroun but for all central Africa countries to improve insecticide resistance management strategies.
Overall, I would definitely recommend the paper for publication with very few minor comments.
Unfortunately as currently presented, there are some editing and lack of information’s which may undermine quality of the paper.
Line 8: What is the affiliation of Charles S Wondji
Line 24: change “Ace-1R mutation” by “G119S mutation”.
Line 28: change “Ace-1R mutation” by “G119S mutation”.
Line 28: change “with” by “in adult field catches collection”
Line 65: change “in” by “some”
Line 65: Put An. gambiae in italic
Line 67: change “ Ace-1 mutation gene in Anopheles gambiae” by G119S mutation in Ace-1 gene of Anopheles gambiae”
Line 80: change “Ace-1R” by “G119S mutation”
Line 89: Why and how did you select this village Bankeng?
Question 1: The title need to be reworded accordingly by involving German cockroach (Blattella germanica L.).
Lines 97 & 113: Put all specie name in italic
Line 106: Why did you store the samples in different conditions?
Line 123: How have you estimated the OR and what was your reference?
Line 127: integrate the accession number of ace-1 gene in vectorbase, it is AGAP001356
Lines 129-136: why there is a difference between number of samples and number of amplicon. If you missed some samples during the sequencing, you adjust your samples number to avoid a confusion.
Line 143: what is the genotype of sample you sequenced for ace-1 duplication identification.
Line 162 -166: what is the result in control tube to be sure that the mortality is due to insecticide? Have you done quality control with susceptible lab strain such as Kisumu to check insecticide paper quality.
Line 196: Adjust the arrows of Figure 3 B properly
Author Response
Reviewer 1
Comments and Suggestions for Authors
This article “The G119S acetylcholinesterase (Ace-1) target site 2 mutation confers carbamate resistance in the major 3 malaria vector Anopheles gambiae from Cameroon: A challenge for the coming IRS implementation” is an extremely interesting research paper describing the actual situation of insecticide resistance in Cameroun which may jeopardize the current vector control strategies. The study was well designed and performed. The authors give a good historical overview on carbamate and organophosphates insecticides resistance in west and Central African and how it may select or migrate in Cameroun.
Moreover, the present article will be of highest interest not only for Cameroun but for all central Africa countries to improve insecticide resistance management strategies.
Overall, I would definitely recommend the paper for publication with very few minor comments.
Unfortunately as currently presented, there are some editing and lack of information’s which may undermine quality of the paper.
Line 8: What is the affiliation of Charles S Wondji
Answer: The affiliation of Charles S. Wondji is now clearly indicated in the manuscript.
Line 24: change “Ace-1R mutation” by “G119S mutation”.
Answer: This suggestion was taken into account and the change was made in the revised manuscript
Line 28: change “Ace-1R mutation” by “G119S mutation”.
Answer: This suggestion was taken into account and the change was made in the revised manuscript
Line 28: change “with” by “in adult field catches collection”
Answer: This suggestion was taken into account and the change was made in the revised manuscript
Line 65: change “in” by “some”
Answer: This suggestion was taken into account and the change was made in the revised manuscript
Line 65: Put An. gambiae in italic
Answer: This suggestion was taken into account and the change was made in the revised manuscript
Line 67: change “ Ace-1 mutation gene in Anopheles gambiae” by G119S mutation in Ace-1 gene of Anopheles gambiae”
Answer: This suggestion was taken into account and the change was made in the revised manuscript
Line 80: change “Ace-1R” by “G119S mutation”
Answer: This suggestion was taken into account and the change was made in the revised manuscript
Line 89: Why and how did you select this village Bankeng?
Answer: Results presented in the submitted manuscript are part of data collected in a framework of a study which aimed to assess the impact on malaria transmission risk of the implantation of irrigated rice fields in the village of BANKENG. So, the selection of Bankeng was not hazardous, but motivated by the implementation of a big agricultural project. More of this information is now added in the Methodology section.
Lines 97 & 113: Put all specie name in italic
Answer: This suggestion was taken into account and the change was made in the revised manuscript
Line 106: Why did you store the samples in different conditions?
Answer: The samples were stored in different conditions following the protocol established in our lab. Indeed, because we performed several transcriptomic analyses, alive mosquitoes are usually kept at -80°C for a good RNA conservation, while dead mosquitoes are stored at – 20°C and are used for DNA extraction.
Line 123: How have you estimated the OR and what was your reference?
Answer: OR were estimated using Vassar stats (http://vassarstats.net/) with a 2x2 contingency table. In each case, the proportion of resistant genotype or allele were compared to the susceptible one. Moreover, for each analysis, the statistical significance was assessed based on Fisher exact probability test. This information is now added in the methods
Line 127: integrate the accession number of ace-1 gene in vectorbase, it is AGAP001356
Answer: This suggestion was taken into account and the accession number of ace-1 one gene was integrated in the revised manuscript.
Lines 129-136: why there is a difference between number of samples and number of amplicon. If you missed some samples during the sequencing, you adjust your samples number to avoid a confusion.
Answer: We totally understand the concern emphasized by the reviewer about the difference between the number of samples used and the number of amplicon obtained. This difference is explained by the fact that some samples did not amplified during the PCR and we decided not to sequence them. To avoid a confusion as recommended by the reviewer, a precision was added in the manuscript as followed: “Out of the 40 samples used, 28 successfully amplified (12 F0 field collected adults, 8 alive and 8 dead after exposure to bendiocarb). These amplicons were sequenced directly using the primers Ex2Agdir1 and Ex4Agrev2 to confirm the presence of the G119S mutation and assess signature of selection at this Ace-1 in this location”
Line 143: what is the genotype of sample you sequenced for ace-1 duplication identification.
Answer: For ace-1 duplication, we used samples from alive mosquitoes 24 hours after exposure to both bendiocarb and propoxur. The selection of the samples was not based to the genotype but to the phenotype of resistance. However, most of them were genotyped as homozygotes resistant.
Line 162 -166: what is the result in control tube to be sure that the mortality is due to insecticide? Have you done quality control with susceptible lab strain such as Kisumu to check insecticide paper quality?
Answer: During our insecticide bioassays no mortality was recordered in the control tubes. Also, the Kisumu strain was used to confirm the control quality of the impregnated papers when first tested. This information is now added in the text.
Line 196: Adjust the arrows of Figure 3 B properly
Answer: This suggestion of the reviewer was taking into account in the revised manuscript
Reviewer 2 Report
I have read this manuscript with pleasure and interest. Elanga-Ndille et al. provide new insights into the spread of a major insecticide resistance mechanism in African populations of Anopheles gambiae. The quality of the core analyses of the paper is high and its conclusions well-supported by the data, and they can be of great use to inform future vector control strategies in Cameroon and neighbouring countries. I recommend its publication after minor revisions (listed below) and the addition of a new analysis.
** METHODS **
L150-152 – Further details regarding the evolutionary analyses are needed. How many sequences were used for each of the analyses (MEGA, DNAsp and haplotype networks)? The authors should also explain which evolutionary model was used in the phylogenetic reconstruction and why was it chosen (did the authors perform any test, e.g. modeltest?).
** RESULTS **
L179-181 – The % of the confidence intervals must be explicitly stated (95% CI?).
L184 – In figure 2, add the raw counts of individuals in each category (either as extra panels or supplementary) so that the readers can easily reproduce your calculations (odds ratios, Fisher’s test if needs be etc.).
L214 – How many sequences were used as input for the haplotype network analysis? All 56 sequences listed in Table 1? Please clarify.
L217 – The authors describe H3 (resistant) haplotype as “ancestral”, but the 119S allele is not ‘ancestral’ in the Anopheles genus. Describing H11 and H3 as the ‘largest’ clusters is less charged.
L219 – It appears that less sequences were used in the phylogenetic analysis than in the haplotype networks (at least judging by the number of nodes that can be seen in the tree). Is that so, and if so, why? Similarly, the authors should clarify why did they use the A. albimanus Ace1 sequence as an outgroup for the phylogeny (here or in the methods). In fact, I have serious doubts that this sequence adds anything to the analysis: it seems to me that the enormous evolutionary distance between this outgroup and the other A. gambiae sequences is confounding the phylogenetic analysis of proper A. gambiae haplotypes, as the branching pattern in Figure 4c is not visible at all. The authors should provide an unrooted tree without extraspecific outgroups.
L223 – The authors should also clarify where do the “Kisumu”, “Ghana” and “Togo” sequences come from (citation, database, etc.) and make the corresponding alignments available as SM (see comments below).
L246 – For figure 5, the authors should provide the same level of detail required for figure 4 as per the previous comments.
** ADDITIONAL RESULTS SECTION **
A joint analysis (haplotype networks & phylogenies) of the data used in figures 4 and 5 would be a key addition to this paper. Specifically, it would clarify whether the susceptible haplotypes found in the duplicated specimens (seen in Figure 5) are more similar to susceptible haplotypes from non-duplicated specimens (seen in Figure 4), or to the resistant and duplicated haplotypes (H6 in Figure 5).
This is an important question because it would provide key information regarding the evolutionary path that led to the emergence of resistance haplotypes combining duplications and 119S. If susceptible haplotypes from Figure 5 derive from the Figure5-H6 cluster, it could mean that they reverted back to the wt 119G allele; which implies that the original duplication occurred on an already mutated haplotype (119S), and that in the past specimens carrying only 119S alleles could have been present. If susceptible haplotypes from Figure 5 are more similar to the susceptible haplotypes from figure 4 (e.g. Figure 4-h3 cluster), it would mean that the 119S mutation first occurred on a duplicated haplotype. Given the fitness costs of 119S alleles, the possibility that 119S could spread in the absence of wt alleles could imply worrying prospects with respect to the spread of resistance to carbamates.
Depending on the quality and clarity of this proposed analysis, these results could be discussed in the context of the studies by Essandoh et al. mentioned in lines 315-319.
** TYPOS **
An. gambiae should be italicised in L65, L97, L113, etc. Please revise.
L179-181 – Nested parentheses should be used as follows: “(regular brackets outside [but square brackets inside] and the sentence ends here)”. In this particular case, however, it would be easier to avoid them by not using “()” for the CI intervals.
** SUPPLEMENTARY FIGURES **
Alongside the PDFs, the alignments should be provided in an accessible format, e.g. as FASTA files. If FASTA is not allowed by the journal’s guidelines, CSV tables are another a possible alternative.
Author Response
Reviewer 2:
Comments and Suggestions for Authors
I have read this manuscript with pleasure and interest. Elanga-Ndille et al. provide new insights into the spread of a major insecticide resistance mechanism in African populations of Anopheles gambiae. The quality of the core analyses of the paper is high and its conclusions well-supported by the data, and they can be of great use to inform future vector control strategies in Cameroon and neighbouring countries. I recommend its publication after minor revisions (listed below) and the addition of a new analysis.
** METHODS **
L150-152 – Further details regarding the evolutionary analyses are needed. How many sequences were used for each of the analyses (MEGA, DNAsp and haplotype networks)? The authors should also explain which evolutionary model was used in the phylogenetic reconstruction and why was it chosen (did the authors perform any test, e.g. modeltest?).
Answer: For the overall analyses (including MEGA, DNAsp and haplotype network), we used 28 sequences. These sequences correspond to the number of samples for which the ace-1 gene was successfully sequenced.
Regarding the model used for the phylogenetic reconstruction, we used the Tamura 3 parameter model. This model was chosen after running the test to get the best model to be used for the analysis. This information is now added in the text.
** RESULTS **
L179-181 – The % of the confidence intervals must be explicitly stated (95% CI?).
Answer: The % of the confidence intervals is now added in the text and it is 95%.
L184 – In figure 2, add the raw counts of individuals in each category (either as extra panels or supplementary) so that the readers can easily reproduce your calculations (odds ratios, Fisher’s test if needs be etc.).
Answer: We thank the reviewer for this suggestion which was taken into account. Raw data used to calculate the OR are now well presented in supplementary file (Supplementary file 2) added to the manuscript.
L214 – How many sequences were used as input for the haplotype network analysis? All 56 sequences listed in Table 1? Please clarify.
Answer: Yes, all the 56 samples listed in Table 1 were used for the haplotype network all coming from the 28 samples sequenced (12 F0, 8 dead and 8 alive) after Phasing.
L217 – The authors describe H3 (resistant) haplotype as “ancestral”, but the 119S allele is not ‘ancestral’ in the Anopheles genus. Describing H11 and H3 as the ‘largest’ clusters is less charged.
Answer: We agreed with the reviewer and his suggestion was considered. So, in the revised manuscript the sentence of line 217 was modified as followed: “The analysis of the haplotype network showed that H3 and H11 were the dominant haplotypes “.
L219 – It appears that less sequences were used in the phylogenetic analysis than in the haplotype networks (at least judging by the number of nodes that can be seen in the tree). Is that so, and if so, why?
Answer: We agree with the reviewer and a new phylogenetic tree is added based on the 56 sequences. The number of sequences used for phylogenetic analysis were exactly the same used for haplotype networks.
Similarly, the authors should clarify why did they use the A. albimanus Ace1 sequence as an outgroup for the phylogeny (here or in the methods). In fact, I have serious doubts that this sequence adds anything to the analysis: it seems to me that the enormous evolutionary distance between this outgroup and the other A. gambiae sequences is confounding the phylogenetic analysis of proper A. gambiae haplotypes, as the branching pattern in Figure 4c is not visible at all. The authors should provide an unrooted tree without extraspecific outgroups.
Answer:, We agree with the reviewer’s observation and we have now removed this An. albimanus sequence from our analysis and figure 4c was modified in the revised manuscript.
L223 – The authors should also clarify where do the “Kisumu”, “Ghana” and “Togo” sequences come from (citation, database, etc.) and make the corresponding alignments available as SM (see comments below).
Answer: We thank the reviewer for this comment. “Ghana” and “Togo” come from a blast we performed on NCBI. Their accession number are as followed: Ghana (KP165343) and Togo (KM875636). Concerning the sequence of “Kisumu” it come from vectorbase (gene ID: AGAP001356). As suggested by the reviewer, all this information was added in the revised manuscript as followed: “Interestingly, the predominant resistant haplotype from F0 and F1 alive mosquitoes was identical to resistant alleles previously detected in Ghana (Accession number: KP165343, NCBI database) and Togo (Accession number:KM875636; NCBI database), in West African (Essandoh et al, 2013)”
L246 – For figure 5, the authors should provide the same level of detail required for figure 4 as per the previous comments.
Answer: This comment was taken into account and changes were made in the revised manuscript as per the answer to the previous comments.
** ADDITIONAL RESULTS SECTION **
A joint analysis (haplotype networks & phylogenies) of the data used in figures 4 and 5 would be a key addition to this paper. Specifically, it would clarify whether the susceptible haplotypes found in the duplicated specimens (seen in Figure 5) are more similar to susceptible haplotypes from non-duplicated specimens (seen in Figure 4), or to the resistant and duplicated haplotypes (H6 in Figure 5).
This is an important question because it would provide key information regarding the evolutionary path that led to the emergence of resistance haplotypes combining duplications and 119S. If susceptible haplotypes from Figure 5 derive from the Figure5-H6 cluster, it could mean that they reverted back to the wt 119G allele; which implies that the original duplication occurred on an already mutated haplotype (119S), and that in the past specimens carrying only 119S alleles could have been present. If susceptible haplotypes from Figure 5 are more similar to the susceptible haplotypes from figure 4 (e.g. Figure 4-h3 cluster), it would mean that the 119S mutation first occurred on a duplicated haplotype. Given the fitness costs of 119S alleles, the possibility that 119S could spread in the absence of wt alleles could imply worrying prospects with respect to the spread of resistance to carbamates.
Depending on the quality and clarity of this proposed analysis, these results could be discussed in the context of the studies by Essandoh et al. mentioned in lines 315-319.
Answer: We found this suggestion of the reviewer very interesting and we totally agree on the importance that the proposed additional result can bring to our work. Taking into account this suggestion, the proposed analysis was performed, and results are now integrated in the revised manuscript (figure 6 and additional file 5). Also, the comment of the results of this analysis was added in the result section of the revised manuscript (lines 263 -278) and in the discussion section (Line 340-348).
From the cloning analysis, six susceptible haplotypes were detected in the duplicated mosquitoes. After the combined analysis, these susceptible haplotypes are H2, H6, H9, H11, H13, and H16. It appears that the susceptible haplotypes from duplicated samples are more similar to susceptible haplotypes from non-duplicated specimens. However, a susceptible haplotype H13-d is nested within a resistant cluster at 2 mutational steps from the dominant resistant haplotype H4 suggesting a possible reversion to the wild type from a resistant haplotype. The resistant haplotypes from duplicated are also all almost similar to those from non-duplicated specimens. This information is now added in the text.
** TYPOS **
An. gambiae should be italicised in L65, L97, L113, etc. Please revise.
Answer: This comment was taken into account and changes were made in the revised manuscript
L179-181 – Nested parentheses should be used as follows: “(regular brackets outside [but square brackets inside] and the sentence ends here)”. In this particular case, however, it would be easier to avoid them by not using “()” for the CI intervals.
Answer: We thank the reviewer for this comment. As he suggested, the “()” used for the CI intervals were removed in the revised manuscript.
** SUPPLEMENTARY FIGURES **
Alongside the PDFs, the alignments should be provided in an accessible format, e.g. as FASTA files. If FASTA is not allowed by the journal’s guidelines, CSV tables are another a possible alternative.
tables are another a possible alternative.
Answer: We understand the concern highlighted by the reviewer and we replaced the pdfs file by the fasta ones. These later were the easiest for us to generate.